# CREAK: A Dataset for Commonsense Reasoning over Entity Knowledge

**Yasumasa Onoe, Michael J.Q. Zhang, Eunsol Choi, Greg Durrett**
The University of Texas at Austin
{yasumasa, mjqzhang, eunsol, gdurrett}@cs.utexas.edu

## Abstract

Most benchmark datasets targeting commonsense reasoning focus on everyday scenarios: physical knowledge like knowing that you could fill a cup under a waterfall [Talmor et al., 2019], social knowledge like bumping into someone is awkward [Sap et al., 2019], and other generic situations. However, there is a rich space of commonsense inferences anchored to knowledge about specific entities: for example, deciding the truthfulness of a claim *Harry Potter can teach classes on how to fly on a broomstick*. Can models learn to combine entity knowledge with commonsense reasoning in this fashion? We introduce CREAK, a testbed for commonsense reasoning about entity knowledge, bridging fact-checking about entities (Harry Potter is a wizard and is skilled at riding a broomstick) with commonsense inferences (if you're good at a skill you can teach others how to do it). Our dataset consists of 13k human-authored English claims about entities that are either true or false, in addition to a small contrast set. Crowdworkers can easily come up with these statements and human performance on the dataset is high (high 90s); we argue that models should be able to blend entity knowledge and commonsense reasoning to do well here. In our experiments, we focus on the closed-book setting and observe that a baseline model finetuned on existing fact verification benchmark struggles on CREAK. Training a model on CREAK improves accuracy by a substantial margin, but still falls short of human performance. Our benchmark provides a unique probe into natural language understanding models, testing both its ability to retrieve facts (e.g., who teaches at the University of Chicago?) and unstated commonsense knowledge (e.g., butlers do not yell at guests).

## 1 Introduction

To understand text, humans use rich background knowledge about the world. Despite the impressive ability of large-scale pretrained models, models often generate sentences that violate a reader's expectations, particularly in terms of common sense. As these models are increasingly employed in settings like generative question answering [Fan et al., 2019, Lewis et al., 2020] and fact verification [Vlachos and Riedel, 2014, Wang, 2017, Thorne et al., 2018], they should exhibit not just commonsense about everyday scenarios (physical, social, etc.), but factual knowledge about entities as well. These concepts overlap in a set of inferences involving entities that we call *entity commonsense*. For example, to recognize that *"Many business owners rely on WordPress to create their websites."* is true requires both knowledge about the entity (WordPress is a website hosting service) and a more nebulous piece of commonsense information (famous products like WordPress are widely used).

We present CREAK, a dataset aiming to evaluate two major desiderata of NLP models: entity understanding and commonsense inference. Figure 1 shows how these concepts interact in examples from CREAK. Building LMs with a stronger ability to perform this type of inference can help make NLP systems more effective and reliable.

35th Conference on Neural Information Processing Systems (NeurIPS 2021) Track on Datasets and Benchmarks.

**Claim:** Harry Potter can teach classes on how to fly on a broomstick. **TRUE**

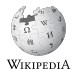 Harry Potter is a wizard … He plays Quidditch while riding on a broomstick. **+** 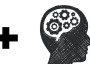 Someone who's good at something can teach it.

**Claim:** One can drive La Jolla to New York City in less than two hours. **FALSE**

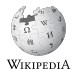 La Jolla is in California. NYC is in New York. **+** 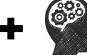 It takes 5h with airplane to fly from California to New York.

**Claim:** François Mitterrand became a Texas Senator in 2001. **FALSE**

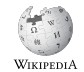 François Mitterrand (26 Oct 1916 – 8 Jan 1996) was a French statesman.

Figure 1: CREAK claims with different reasoning types. Datasets like FEVER focus on retrieval (as in the last case); our dataset also features many claims that involve both retrieval and also commonsense reasoning (54% of the data according our manual study in Section 3.3). Note that the examples in this figure do not explicitly outline all the reasoning steps needed for a system (e.g., we assume that Harry Potter is good at riding on a broomstick since he is a member of a Quidditch team.).

Our dataset consists of 13k English claims covering 2.7k entities, each labeled as true or false. Each claim is generated by a crowdworker based on a Wikipedia entity, which can be named entities (e.g., John Dewey), common nouns (e.g., penguins), and abstract concepts (e.g., freedom of speech). Our lightweight task design provides annotators with a set of popular entity topics, and by not including explicit evidence documents to copy text from, annotators are encouraged to create examples fully from scratch. This results in sentences where annotators combine their knowledge about entities with common sense to generate claims. Even without resorting to adversarial filtering, which artificially biases a dataset against existing model checkpoints [Bowman and Dahl, 2021], we find our annotation protocol leads to challenging claims for existing models. We provide in-depth analysis on what makes our dataset uniquely challenging: for example, 18% of claims in CREAK contain quantifiers (e.g., enough, always, rarely etc.) that necessitate subtle commonse reasoning, compared to existing fact verification datasets [Thorne et al., 2018] where only 5% of claims contain the quantifiers.

Asking crowdworkers to generate free-form sentences can introduce dataset artifacts [Gururangan et al., 2018, Geva et al., 2019]. We carefully examine such artifacts in our datasets using quantitative tools [Swayamdipta et al., 2020, Gardner et al., 2020] as well as qualitative inspection. We also provide a small set of expert-written contrast examples [Kaushik et al., 2019, Gardner et al., 2020] which pair true and false claims sharing almost identical context.

To establish an initial performance level on CREAK, we evaluate state-of-the-art pre-trained language models [Liu et al., 2019, Raffel et al., 2020]. Our experiments shows that CREAK is challenging even for a large model, with a gap between model and human accuracy of 10 points on the development set and about 27 points on the contrast set for the largest model. Moreover, the model trained on CREAK outperforms the model trained on other claim verification datasets [Thorne et al., 2018, Eisenschlos et al., 2021, Park et al., 2021], suggesting that CREAK tests different reasoning capabilities compared to existing datasets. We further characterize the performance based on model size, entity type, and the whether external knowledge is used. Our analysis supports that to achieve high performance on our dataset, models should possess not only entity knowledge but also complex reasoning skills. The code and data are publicly available at `https://www.cs.utexas.edu/~yasumasa/creak`.

## 2 Related Work

**Claim Verification** Our task is formulated as claim verification, which has seen increasing work in recent years. The largest claim verification dataset, FEVER [Thorne et al., 2018], has claims designed to be verifiable with a passage from English Wikipedia and typically covers simple facts such as attributes and records (e.g., *"Benjamin Franklin was a person,"* and *"Spider-Man 2 was released in 2004."*). In fact, 58% of FEVER claims contain a simple copular verb (*is, are, was, were*) and many claims contain a definition. Prior work [Eisenschlos et al., 2021] observed high lexical overlap between claims and corresponding entity definitions in Wikipedia, and collected more complex claims

using a human-in-the-loop adversarial approach. Similarly, recent work [Park et al., 2021] derives a challenging verification dataset from ambiguous questions [Min et al., 2020] and their different interpretations. However, both datasets focus on a retrieval setting where there is a single paragraph in Wikipedia from which the claim can be easily verified. In contrast, our dataset contains claims where it is not easy to find a single paragraph that can verify them, testing models' intrinsic abilities.

**Question Answering**    Question answering and claim verification are closely related, particularly when it comes to binary questions [Clark et al., 2019]. Our dataset is purposely constructed to go beyond basic factoid information like that tested in open QA benchmarks like NaturalQuestions [Kwiatkowski et al., 2019] and focuses on information that is less likely to have textual support on the web. The recently proposed StrategyQA dataset [Geva et al., 2021] which contains binary questions requiring implicit reasoning that goes beyond evidence retrieval (e.g., *"would it be common to find a penguin in Miami?"*) captures a similar type of reasoning and knowledge as in our work. However, our annotation process does not require authoring strategies, allowing us to scale to a larger dataset (13K vs. 2.8K) while capturing a wide range of inference types. Finally, some QA datasets have been adapted for evaluating differentiable commonsense reasoning models [Lin et al., 2021], but these benchmarks still test very different knowledge from ours.

**Commonsense Reasoning**    Commonsense reasoning tasks [Levesque et al., 2011, Zellers et al., 2018, Talmor et al., 2019, Lourie et al., 2021, inter alia] evaluate models' reasoning skills in the physical world, with reporting bias being a principal challenge [Gordon and Van Durme, 2013]. Yet, most datasets assume hypothetical environments and do not address real-world entities. Our work relates to judging plausibility of events [Forbes and Choi, 2017, Wang et al., 2018], closely tied to inferences accessible from feature norms [McRae et al., 2005], but again these past efforts do not focus on judgments around specific entities.

**Knowledge Probing**    The LAMA benchmark [Petroni et al., 2019] was proposed to query factual knowledge covered in language models. Our dataset also covers such factual knowledge but also requires commonsense reasoning capabilities. Our work also creates a moderately sized training dataset. Other datasets in the KILT benchmark [Petroni et al., 2020], an aggregate suite focusing on knowledge intensive tasks, are more focused on recognizing entities and relations, "low-level" factual knowledge which does not require the kinds of commonsense inferences in our dataset. Another recent commonsense-focused dataset [Lin et al., 2020], focuses on probing numeric claims.

## 3  CREAK

### 3.1  Task Definition

**Problem Scope**    Our benchmark covers claims that are typically quite easy for humans to verify but challenging for language models. We focus on factual claims about real-world entities, but our claims are more complex than existing fact verification examples which tend to state relatively simple facts (i.e., definitive sentences, *X is a Y*, or sentences expressing simple relations, like *X is CEO of Z*). To the extent possible, we avoid information that is obscure or requires computation, such as asking about the time between two arbitrary events or how many copies of an album were sold, which test either retrieval or memorization rather than commonsense reasoning. We found that our claims can often be verified with minimum knowledge of the entities combined with common sense (i.e., you can guess the answer accurately even if you do not know the entity very well).[1] We argue that this knowledge is what pre-trained LMs should possess about moderately well-known entities after seeing a few occurrences of them during pre-training. Therefore, our claims should be solvable in the closed-book setting where we can purely evaluate LMs' commonsense reasoning skills, isolated from the performance of retrieval models.

We formally define the CREAK  task as follows. Given a single sentence claim $c$ containing at least one entity mention, the task is to assign a binary label $y \in \{\texttt{TRUE}, \texttt{FALSE}\}$ indicating whether the claim is true or false. Dataset statistics can be found in Table 1.

---

[1]During our validation, we could confidently judge about 30-50% of claims without searching the web.

Table 1: Data statistics of CREAK.

| Split | # Claims | | | Average Length (# tokens) | # Unique Entities | Vocab Size |
|---|---|---|---|---|---|---|
| | Total | True | False | | | |
| Train | 10,176 | 5,088 | 5,088 | 10.8 | 2,096 | 19,006 |
| Dev | 1,371 | 691 | 680 | 9.7 | 531 | 4,520 |
| Test | 1,371 | 707 | 664 | 9.9 | 538 | 4,620 |
| Test (Contrast) | 500 | 250 | 250 | 10.0 | 226 | 1,596 |

**Dataset Properties** Our dataset has following key properties. The claims are **diverse** covering various types of entities: they are written by 684 distinct crowdworkers[2] only based on the entity names and their minimal information. We rarely find lexical overlap between the claims and publicly available knowledge sources (e.g., the first paragraph of English Wikipedia). As a result, the claims contain a variety of reasoning types, but nevertheless are **typically not subjective** and **easily verifiable**. As discussed in Section 3.3, a majority of our examples do **involve a combination of commonsense reasoning and knowledge**. Finally, Sections 3.3 and 4 show that the dataset is **relatively robust to spurious correlations**.

## 3.2 Data Collection

We collect our data on the Amazon Mechanical Turk platform (see examples in Table 2). Open-ended text generation is challenging to crowdsource, so we take several steps in our task design to ensure quality. First, we ask crowdworkers to write down the reason why the generated claim is true or false; although past work observes that this does not improve example quality in isolation [Nangia et al., 2021], we found it helpful for our task, and it additionally helped us spot workers who misunderstood the task. To keep the sentences natural, we use a minimal set of requirements and encourage crowdworkers to produce creative and diverse sentences. One key requirement is to use action verbs instead of copula, which prevent crowdworkers from writing simple definitive sentences. See Appendix A for more details about the annotation instructions.

We do not take a model-in-the-loop approach [Zellers et al., 2018, 2019, Nie et al., 2020, Bras et al., 2020] during data collection in order to keep our dataset organic, meaning that sentences preserve the original data distribution given by annotators. Therefore, this benchmark does not favor or disfavor particular LM checkpoints, providing a fair and comparable testbed [Bowman and Dahl, 2021].

**Seed Entities Curation** Entity selection plays a crucial role in this task, since authoring sentences is a much easier task if a crowdworker is familiar with the entity. We take two steps to enable crowdworkers to focus on known entities. First, we use the entity list created by Geva et al. [2021] as part of StrategyQA, which aligns with our needs; the authors select entities based on some popularity measures such as the number of contributors and the number of backward links from other pages. Second, we present five entities to each annotator and let them pick from that set of five when authoring their sentences. We manually inspect the seed entities to maintain the diversity of the types of entities so that the generated claims cover diverse topics (e.g., we want to avoid too many location entities that occur in English Wikipedia frequently). We finally obtain 6.4k entities after this process.

We split the seed entity list into two parts; one for the training instances and one for the development and test instances. In both sets, roughly 80% of entities are named entities. The 5 most popular entities in the train set are *Sloth*, *Giraffe*, *George Orwell*, *50 Cent*, *Mattel*. In the development and test sets, *Butterfly*, *Ray Charles*, *Whole Foods Market*, *Internet troll*, and *Bigfoot* are the top 5 popular entities. As can be seen, crowdworkers prefer to select relatively common entities.

**Quality Control** We split the data curation into two separate tasks such that no annotator contributed to both training and evaluation datasets. This mitigates the issue of learning to model the behavior of specific annotators [Geva et al., 2019] and annotation artifacts from annotator developing a template (e.g., ENTITY created ENTITY) across many instances. In total, CREAK is created by a large number of annotators: 153 crowdworkers annotated the development and test instances, and 531 crowdworkers worked on the training instances. We also use disjoint sets of entities between

---

[2]We limit the number of claims that a single crowdworker can generate (no more than 7% of any split).

Table 2: CREAK examples selected from the development set. Boldface spans indicate the seed entities presented to crowdworkers.

| Claim | Label |
|---|---|
| In **robotics**, some robots are autonomous, while others need human assistance to operate properly. The **Curse of the Bambino** began when Babe Ruth was traded to the Yankees. In today's world **dessert** comes at any time, not just after dinner. **Nancy Pelosi** followed Paul Ryan as Speaker of The House. | TRUE |
| You must have a university degree in order to be able to get hired as a **civil engineer**. The **Pepsi Center** sprays its audience with Pepsi after every point scored. Driving cars at the **Las Vegas Motor Speedway** is only for the public. People use **spaghetti** to tie items together. | FALSE |

training and dev/test data so a model trained on the dataset is not simply learning properties of the entities under discussion here. We discuss more in Section 3.3.

During the annotation process, we monitored the sentence quality and barred crowdworkers who repeatedly produced low-quality sentences or examples following a single pattern. We then inspected the examples included in our evaluation dataset. During the inspection, we found some claims that are subjective, ambiguous, or non-factual (see Appendix B). These errors potentially lower the human performance on the development and test sets. Since automatically detecting these errors is non-trivial, the authors manually filtered all claims in the evaluation dataset. This process removed roughly 18% of crowdsourced claims. This process was crucial for very high human performance (99% majority human performance), as we will see in the experiments.

**Contrast Set**  The authors of the paper created a small subset of contrastive examples [Kaushik et al., 2019, Gardner et al., 2020]. We select 250 seed claims from the evaluation set, then annotate true and false claims based on the seed claims by applying minimal modification (e.g., replacing a word with a similar one that changes the truth of the claim). Examples can be found in Appendix B.

### 3.3 Dataset Analysis

In this section, we examine the quality of our dataset. We first manually examine what types of reasoning are required to verify our claims. Then, we study potential lexical and syntactic artifacts in human-generated claims through statistical tests and training dynamics to identify word-level artifacts and learnability of the training instances.

**Manual Analysis of Reasoning Types**  We manually validate whether the CREAK claims truly require both knowledge and commonsense. We classify reasoning types into three categories: 1) retrieval, 2) common sense, and 3) a mix of retrieval and common sense. These distinctions are somewhat subjective based on the background knowledge of the reader (i.e., is it common sense that NYC is a major city?); we use our own judgments as authors. The first category, retrieval, asks simple facts about entities which can be found in some knowledge sources such as English Wikipedia (e.g., *The Harry Potter series originally began with the books.*). The second category, common sense, requires more complex reasoning but are verifiable with the basic knowledge of the entities (e.g., *Only trumpet players can perform a solo.*). The third category is a mix of retrieval and common sense, meaning that it involves some degree of retrieval and commonsense reasoning. For example, the claim *One can drive from La Jolla to New York City in less than two hours.* requires knowing the locations of La Jolla and New York City (retrieval) and reasoning about driving times (common sense). We randomly sample 100 claims from the evaluation instances and classify them into the three categories. The proportion of the retrieval, common sense, and a mix of the two categories is 18%, 28%, and 54% respectively. You can find examples for each reasoning type in Appendix B.

**Dataset Artifacts**  Past work on natural language inference has noted that "artifacts," or spurious correlations with surface properties of text, may arise during the annotation process [Gururangan et al., 2018, Poliak et al., 2018]. The low performance of a bag-of-words model in our setting (see Table 3) gives some confidence that such correlations are not a dominant factor in performance on our data, but we undertake quantitative analysis to explore this further.

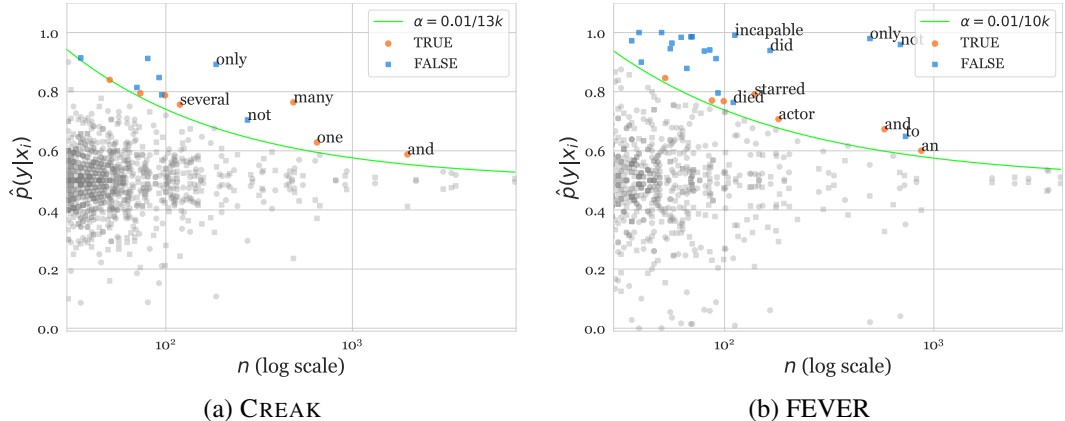

(a) CREAK         (b) FEVER

Figure 2: Artifact statistics of CREAK and FEVER train sets. Words (colored dots) above the green line have detectable correlation with class labels. CREAK contains relatively fewer artifacts, with low severity and frequency.

We identify the word-level artifacts in CREAK by computing the artifact statistics described in Gardner et al. [2021]. These statistics tell us, given a balanced dataset, if some words are highly correlated with either true or false claims in a way that a model can exploit. This boils down to a one-side binomial hypothesis test with the null hypothesis $p(y|x_i) = 0.5$, where $y \in \{\texttt{TRUE}, \texttt{FALSE}\}$ is a label and $x_i$ is a word in the vocabulary. We first count the occurrence of all words[3] in CREAK . For each word $x_i$ that appears in the $n_i$ claims, we count the number of the target label $y$ in the $n_i$ claims. We estimate $p(y|x_i)$ with the observed probability $\hat{p}(y|x_i)$, which is given by a fraction of the count of $y$ over $n_i$. Following Gardner et al. [2021], we then compute a $z$-statistic and reject/accept the null hypothesis using $\alpha = 0.01$ with the Bonferroni correction.

Figure 2 plots the word counts $n_i$ ($x$-axis) against the observed probability $\hat{p}(y|x_i)$ ($y$-axis) for CREAK and FEVER dataset. We additionally draw the curve that represents the corresponding probability of $\alpha = 0.01/13k$ (for CREAK) and $\alpha = 0.01/10k$ (for FEVER) at each $n_i$. Any words above this line are considered to be artifacts in the dataset. We find 14 words (out of 13k words in the vocabulary) sit above the line. We label the most frequent words in the plot. Surprisingly, *and* ($n = 1973$) is the most frequent artifact that signals the true label, followed by some quantifiers (*many*, $n = 483$, and *several*, $n = 119$). *not* ($n = 274$) and *only* ($n = 186$) suggest the false label in both datasets. Overall, CREAK contains relatively few artifacts, and they do not impact the data quality significantly since their frequency is not very high. We observe fewer artifacts compared to FEVER dataset (14 words vs. 28 words above the threshold).

**Training Dynamics**     We analyze training dynamics using the framework proposed by Swayamdipta et al. [2020]. The training dynamics of a training instance are defined by *confidence* and *variability*,the mean and the standard deviation of model predictions (probability) on the gold label over training epochs. Additionally, *correctness* is computed by the number of times a training instance is correctly predicted over the number of epochs. Figure 3 shows the histograms of those measurements[4] for CREAK (10k instances) and FEVER (105k instances). We use ROBERTA Large[5] for all experiments. In the *confidence* plots, CREAK has a fatter distribution (i.e., certain instances get low probability on their gold labels) compared to FEVER's skewed distribution where the majority of instances get very high probability (e.g., > 0.9) on the gold labels. CREAK's *variability* histogram is nearly bell-shaped while FEVER's histogram skews towards zero. As can be seen in the *correctness* plots, some training instances in CREAK are not always predicted correctly during training, as its distribution suggests. However, the most of training instances of FEVER are correctly predicted consistently through the training epochs. By aggregating these observations, we hypothesize that CREAK contains training instances with different difficulty levels compared to FEVER.

---

[3] We drop punctuation and lower all words.

[4] All values are normalized between 0 and 1 then bucketed into subgroups.

[5] Following the suggestions by Swayamdipta et al. [2020], we train models with early stopping with patience = 3, resulting in 7 epochs for CREAK and 6 epochs for FEVER.

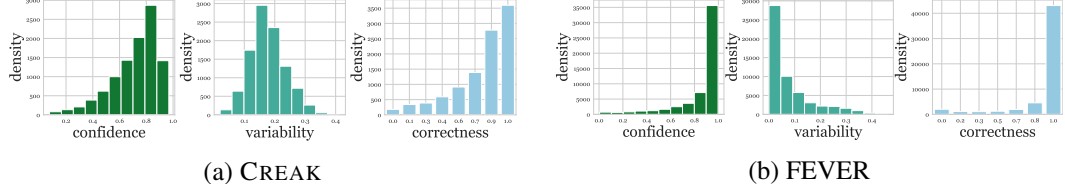

(a) CREAK                                      (b) FEVER

Figure 3: Training dynamics for CREAK and FEVER train sets. This figure shows histograms of CREAK or FEVER training instances bucketed by *confidence* (mean), *variability* (std.), or *correctness*. On all three measures, CREAK shows fatter distributions compared to FEVER, implying that CREAK consists of instances with different difficulties.

# 4 Experiments

We focus on the closed-book setting where models are ask to make decisions based solely on claims without any additional retrieved evidence. To see if existing claim verification datasets provide entity commonsense supervision, we train claim-only baseline models on FEVER [Thorne et al., 2018], FAVIQ [Park et al., 2021], and FOOLMETWICE (FM2) [Eisenschlos et al., 2021] and then evaluate them on CREAK . Next, we train models on the CREAK training set and measure the improvements over the baselines. We also investigate the impacts of model sizes and external knowledge.

## 4.1 Experimental Setup

We investigate three training data settings. In the *Zero-Shot* setting, we train models on the train sets of FEVER $_{KILT}$, FAVIQ-A, FAVIQ-R,[6] and FOOLMETWICE (FM2). In the *In-Domain* setting, we train models on the CREAK train set in a standard fashion. The *Finetuning* setting means that we train models on FEVER and then further finetune on CREAK.

We evaluate all models on the CREAK balanced development, test, and contrast sets and report accuracy. As we discussed in Section 3, these evaluation sets use distinct entities from the train set and are authored by a different set of crowdworkers.

## 4.2 Comparison Systems

**Closed-book (Claim-only) Models**    In what we consider our standard setting, these models take a claim as input and predict if the claim is true or false. We use a RoBERTa encoder [Liu et al., 2019] with a MLP classifier for baseline models: ROBERTA $_{Large}$ and ROBERTA $_{Base}$. We also train SVM with TF-IDF, which gives a linear baseline using far fewer parameters than the LM-based models. We further employ T5-3B to see if more parameters help to learn the complex reasoning in CREAK.

**Retrieval-based Models**    These models are augmented with knowledge retrieved from Wikipedia. We feed a claim and $k$ retrieved passages to a model, which can use the information in the passages to influence the decision. We use Dense Passage Retrieval (DPR) [Karpukhin et al., 2020],[7] a dual-encoder based model, as a retriever and English Wikipedia as a knowledge base. Specifically, we use the DPR model trained on the KILT benchmark, which includes FEVER. We use this configuration for the open-book experiments, where we finetune models on our training set as well as on FEVER. For a claim classifier, we use the ROBERTA $_{Large}$ model and denote this retrieval-based model as ROBERTA $_{Large-DPR}$. We retrieve $k = 3$ passages for all experiments.

**Human Performance**    To estimate human performance on the development set, we sample 100 examples and predict the corresponding labels. For the contrast set, three of the authors predict labels for claims that they did not annotate. We report the averaged human accuracy and the ensemble accuracy which we use the majority label as the final prediction to computer human performance. The temporal mismatch between LMs (e.g., RoBERTa was trained on the December 2018 EN Wikipedia dump.) and CREAK (collected claims between June and August 2021) can disadvantage models

---

[6]The FAVIQ benchmark consists of two datasets based on the same source QA dataset.

[7]DPR is licensed under CC BY-NC 4.0

Table 3: Performance of closed-book approaches on CREAK. Transfer results from prior datasets show that our dataset is distnct from these. Larger models trained with in-domain data perform the best out of all models we consider, but still lag behind human performance.

| | Model | #Params | Training Data | | Accuracy | | |
| | | | Type | Size | Dev | Test | Contrast |
|---|---|---|---|---|---|---|---|
| | Majority Label | – | – | – | 51.6 | 51.6 | 50.0 |
| *Zero-Shot* | ROBERTA $_{Large}$ | 355M | FAVIQ-R | 141k | 49.6 | 48.4 | 50.0 |
| | ROBERTA $_{Large}$ | 355M | FAVIQ-A | 17k | 52.3 | 52.6 | 51.4 |
| | ROBERTA $_{Large}$ | 355M | FM2 | 10k | 59.2 | 58.2 | 54.4 |
| | ROBERTA $_{Large}$ | 355M | FEVER $_{KILT}$ | 105k | 69.6 | 70.2 | 59.6 |
| | T5-3B | 3B | FEVER $_{KILT}$ | 105k | 68.7 | 70.8 | 62.8 |
| *In-Domain* | SVM + TF-IDF | 13k | CREAK | 10k | 60.2 | 60.3 | 53.8 |
| | ROBERTA $_{Base}$ | 125M | CREAK | 10k | 72.2 | 71.6 | 58.8 |
| | ROBERTA $_{Large}$ | 355M | CREAK | 10k | 80.6 | 80.3 | 66.6 |
| | T5-3B | 3B | CREAK | 10k | **85.6** | **85.1** | **72.6** |
| *Finetuning* | ROBERTA $_{Large}$ | 355M | FEV → CREAK | 115k | 80.5 | 81.1 | 68.6 |
| | Human (averaged) | – | – | – | 96.3 | – | 92.2 |
| | Human (ensemble) | – | – | – | 99.0 | – | 99.0 |

[Zhan and Choi, 2021]. However, when we manually inspect 100 examples, none of the truth values would have changed in this time period and nearly all of the entities were already notable as of 2018.[8]

## 5  Results and Discussion

Table 3 presents our main experimental results for closed book systems, and Table 4 presents results for retrieval augmented approaches. We observe that all baseline models fall behind our estimated human performance by a substantial margin.

**Transfer from existing datasets**  The *zero-shot* block of Table 3 compares performance of RoBERTa models trained on four prior claim verification datasets. The models trained on FAVIQ-R and FAVIQ-A perform similarly with the majority label baseline. The model trained on FM2 shows better performance than the FAVIQ models, but the accuracy is still very low. We see much improved transfer from FEVER $_{KILT}$ dataset, reaching an accuracy of 70%. Although designed to be more challenging than FEVER, FAVIQ and FM2 may result in models that transfer less well because these datasets are more dependent on retrieving specific passages to judge claims, containing fewer claims resolvable with commonsense reasoning. Additionally, T5-3B trained on FEVER $_{KILT}$ performs similarly to ROBERTA $_{Large}$ although it is 8 times larger, suggesting that FEVER $_{KILT}$ is bounded in terms of how useful it can be for CREAK .

In the *Finetuning* block of Table 3, we report the performance of ROBERTA $_{Large}$ first trained on FEVER $_{KILT}$ and then on CREAK. Compared to ROBERTA $_{Large}$ trained only CREAK, additional pre-training does not bring meaningful gains.

**Are larger models better?**  The *In-Domain* block of Table 3 lists performance by models with different sizes ranging from 13k to 3B parameters. All models are trained on CREAK. ROBERTA $_{Base}$, outperforms SVM with TF-IDF by 11 points on the test set, suggesting that a larger, knowledge-rich model can do better. But its advantage shrinks on the contrast set, only gaining 5 points over SVM with TF-IDF. A larger model ROBERTA $_{Large}$, 355M parameters, further improves the performance, and this trend continues to an even larger model T5-3B, which outperforms ROBERTA $_{Large}$ by 5 points on the test set and 6 points on the contrast set. T5-3B achieves the highest accuracy in the closed-book setting. Given how the contrast set was constructed, the fact that higher-capacity models work better suggests that having more entity knowledge is a key to doing better on this task.

---

[8]The only example that arguably relies on more recent knowledge is *"Captain Marvel shows a woman using her swimming powers to save people"*, since the popular Captain Marvel film was released in 2019. However, the character of Captain Marvel did exist before, so this information was within reach from the RoBERTa model.

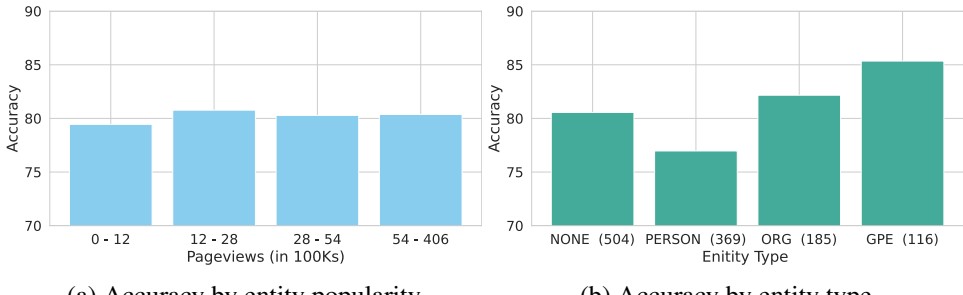

(a) Accuracy by entity popularity      (b) Accuracy by entity type

Figure 4: Performance breakdown on partitions of the development set, split by entity popularity and type, using the four most common entity types, which comprise 86% of examples (includng "NONE" for non-named entities).

Table 4: Performance of retrieval-augmented approaches on CREAK. Large models retrieving from Wikipedia can do better, although the performance on the contrast set is still low.

| | Model | #Params | Training Data | | Accuracy | | |
| | | | Type | Size | Dev | Test | Contrast |
|---|---|---|---|---|---|---|---|
| | Majority Label | – | – | – | 51.6 | 51.6 | 50.0 |
| *Zero-Shot* | ROBERTA Large+DPR | 575M | FEVER KILT | 105k | 79.5 | 80.7 | 71.2 |
| *In-Domain* | ROBERTA Large | 355M | CREAK ENT_LESS | 10k | 66.5 | 67.8 | 56.8 |
| | ROBERTA Large+DPR | 575M | CREAK | 10k | 84.9 | 84.3 | 73.4 |
| *Finetuning* | ROBERTA Large+DPR | 575M | FEV → CREAK | 115k | **88.7** | **86.8** | **76.0** |
| | Human (averaged) | – | – | – | 96.3 | – | 92.2 |
| | Human (ensemble) | – | – | – | 99.0 | – | 99.0 |

**Performance breakdown by entity types** We examine whether models are better equipped at verify claims about different entities depending on their popularity and type, as given by an NER tagger. We use ROBERTA Large as an in-domain baseline model for this analysis. To compare entity popularity, we partition our dataset into equally sized quartiles based on total number of views the entity's Wikipedia page has received since Jan. 1, 2016. For entity types, we use an off-the-shelf NER tagger from spaCy [Honnibal et al., 2020] to group examples by the entity type. In Figure 4, we plot the performance on each partition of our dataset. We observe that the model performs comparably regardless of entity popularity, partially because we sampled from popular entities, and that entity type has a greater affect on accuracy.

**Retrieval-based models with external knowledge** To investigate the importance of entity knowledge in CREAK , we experiment in two additional settings. First, to confirm that entities are important, we experiment with the closed-book setting where all entities are dropped from the claims; this data is denoted as CREAK ENT_LESS. Second, we explore the retrieval setting, where we append three English Wikipedia passages retrieved by DPR to the claims. Similar to the main experiments, we use three data settings: *Zero-Shot*, *In-Domain*, and *Finetuning*.

Table 4 shows the results of all models. ROBERTA Large trained on CREAK ENT_LESS loses more than 10 points compared to the model trained on the standard CREAK training set (*In-Domain* ROBERTA Large in Table 3). This shows that seeing the entity mention in question is important. For open-book models, we again see that *In-Domain* models are better than *Zero-Shot* models. One distinction from the closed-book setting is that the additional finetuning on FEVER KILT improves performance. If we compare the *In-Domain* model from the closed and retrieval settings, the additional passages bring 4 points of improvement. Although adding more entity knowledge improves performance on CREAK, there is still a gap from the human performance, particularly on the contrast set. This shows that there are some facts immediately retrievable from Wikipedia; however, our analysis of the dataset also shows that significant additional reasoning is required as well. Moreover, we believe that this kind of knowledge should be accessible to models in a closed-book way, as annotators were able to create these examples without consulting Wikipedia or other knowledge sources.

# 6   Conclusion

We have presented a dataset CREAK of binary claims involving "entity commonsense," a combination of entity knowledge and commonsense reasoning. This dataset is useful both as a training set for instilling this kind of reasoning into models as well as a test set for probing whether models can recognize factually incorrect statements about entities. We believe this can be a useful proving ground for models infused with entity knowledge (e.g., entities-as-experts [Févry et al., 2020] or interpretable entity embeddings [Onoe and Durrett, 2020]) and contribute to development of these techniques.

**Limitations and Ethical Concerns**   We emphasize that our dataset is not intended for training general fact-checking models; we do not support large-scale deployment of models trained on CREAK for this purpose. Furthermore, while we have tried to measure artifacts in this dataset and found them to be minimal, our claims are artificially generated and the nature of the dataset can differ significantly from claims naturally occurring in social media or web. Large language models fine-tuned on our dataset may preserve biases learned from the web text during pre-training or biases of our annotators and make biased judgments as a result. See the datasheet in the Supplementary Material for more information about specific harms that could arise from this dataset.

# Acknowledgments

This work was partially supported by NSF Grant IIS-1814522; this support included funding for the data annotation. The authors would like to thank Mor Geva for providing the raw list of entities used in StrategyQA, as well as the Mechanical Turk annotators who participated in our task.

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
