# OpenReview forum: "CREAK: A Dataset for Commonsense Reasoning over Entity Knowledge"
_NeurIPS.cc/2021/Track/Datasets_and_Benchmarks/Round2 — NeurIPS 2021 Datasets and Benchmarks Track (Round 2)_

### Official Review · Reviewer_VoQi · 2021-09-19
**Timely work with a number of limitations**

**Rating:** 6
**Confidence:** 5

**Strengths:**

This work is well positioned as part of a growing movement of examination of the type of knowledge stored in large-scale LMs. The main appeal is use of commonsense knowledge during claim construction which isn't easily captured by existing pre-trained models.

**Weaknesses:**

In addition to its relatively small size (compared e.g. to FEVER, LAMA), the main limitation is the degree of control over the elements of the dataset, namely the amount and type of commonsense reasoning needed for each claim, and the lack of any inspection/explanation mechanisms (e.g. evidence retrieval) - indeed the suggested vision of Figure 1, where a commonsense statement is appended to relational knowledge as evidence for a claim is not materialised in the paper (see Correctness section for more details). Finally, as the authors admit, as the claims in this dataset are artificial, it's not possible to use this dataset to examine how large-scale LMs will behave against real-world claims.

**Additional Feedback:**

No additional feedback.

**Clarity:**

The paper is generally very well written. A few minor issues:
- Missing closing bracket at the end of footnote 2
- "minimal information" (line 123) isn't defined
- Unnecessary use of "the authors" in paragraphs where "we" is also used in the rest of the paper (lines 170, 173) - especially the mention in line 265 ("we [...] ask the authors of this paper").

**Correctness:**

The manual analysis provided in Section 3.3 (classification into reasoning types) displays two methodological flaws with this work. As the authors admit, distinguishing between the different reasoning types (relational knowledge, commonsense or a mixture) is "somewhat subjective based on the background knowledge of the reader". As there is no attempt at an operational definition of what counts as commonsense knowledge and what counts as world (relational knowledge), this leaves any attempt at developing an automated solution without a solid foundation - optimising rather against the subjective judgements of annotators and adjudicators. For example, I find it very hard to believe that knowing how long it takes to fly from California (anywhere in CA?) to New York is commonsense (at least it's not if you're not American). Going beyond an operational definition however, there is the problem of defining the level of inference required to solve these claims. If we approach this task with a formal reasoning perspective, examples like the first claim in Figure 1 don't work (the commonsense part says that someone needs to be good at something before teaching it whereas we are never told that HP is good at riding on a broomstick). Another limitation is that the authors couldn't (or didn't) control for the distribution among the different types of reasoning required.

**Documentation:**

The creation and distribution of the dataset is thoroughly discussed in the paper.

**Ethics:**

No additional ethical concerns.

**Relation To Prior Work:**

There is good discussion of previous work and the comparison with FEVER is thorough and interesting (especially the use of the recently introduced artefact statistics and training dynamics analyses).

**Summary And Contributions:**

The paper presents a new dataset for testing commonsense reasoning, based on factual statements about entities. In contrast to previous fact-checking datasets (e.g. FEVER), the emphasis is on the commonsense aspect which is combined with relational statements about popular entities. Despite this, the results sugest that main driver of performance is still memorisation of information on the part of large LMs (the larger the better).

---

> ### Author Response · Authors · 2021-09-29
> **Author Response**
>
> Thanks for the constructive feedback! We address the individual points below.
>
> > _its relatively small size_
>
> We agree that the dataset size is relatively small compared to some large-scale fact verification datasets such as FEVER. Those large-scale datasets can be efficiently collected by providing seed sentences from English Wikipedia. However, this protocol design also results in low-quality claims. For example, many of FEVER claims are generated from the first line of Wikipedia pages, and thus they have similar sentence patterns (e.g., X is Y.). Eisenschlos et al. (2021) point out the same issue in FEVER and create the FoolMeTwice (FM2) dataset, in which the authors create more complex claims using a human-in-the-loop adversarial approach. The size of FM2 is similar to ours (10k train, 1.2k dev, 1.4k test). These dev and test sets are still large enough to tease apart differences between systems.
>
> In our case, we ask crowdworkers to generate free-form sentences just based on basic information about entities (we provide entity names and types only). Since crowdworkers need to come up with claims from scratch, this process is more time-consuming. We decided on our dataset size by considering the tradeoff between the number of examples and cost.
>
>
> > _the degree of control over the elements of the dataset, namely the amount and type of commonsense reasoning needed for each claim_
>
> In this work, precisely controlling the amount and type of commonsense reasoning is not our goal. As we describe in Section 3.1, we aim to collect diverse claims that cover a range of entities combined with many reasoning patterns. We draw a contrast between our work and StrategyQA: that dataset is constructed using a much more heavyweight and expensive annotation protocol, yielding a smaller dataset size.
>
> In our pilot study, we found that reasoning patterns used by crowdworkers are highly varied and not easy to characterize. For example:
>
> - Right/wrong affiliation/classification
> - Similar sounds (e.g., dessert and desert)
> - Matching/mismatching sense organs and their inputs (e.g., measuring a foot by ears)
> - Malfunctioning system (e.g., human without a heart)
> - Right/wrong characteristic (e.g., helium is the lightest element)
> - Inability (e.g., a human can safely eat plutonium, Justin Bieber performed at Abraham Lincoln's inauguration)
>     - Physically impossible
>     - Geographically impossible
>     - Temporally impossible
> - Imaginary event
> - Negating a true statement
> - Matching/mismatching position/location (e.g., pancreas is right above the brain)
> - Matching/mismatching tradition (e.g., camels are common pets in US)
> - Matching/mismatching temporal information (e.g., Easter in October)
> - Matching/mismatching functionality
> - Matching/mismatching purpose
> - Outdated information (i.e., not true anymore)
>
> We realized that classifying claims into well-defined categories was not tractable, and also that forcing crowdworkers to conform to such categories would limit the diversity of reasoning types. Thus, we decided to take a top-down approach where we ask crowdworkers to generate diverse claims first and take them as they are (we filtered unusable claims -- see Appendix B, Table 5). To ensure quality, we finetuned annotation requirements (e.g., no simple "X is Y" type statements, minimum length etc.) through multiple rounds of pilot experiments. To ensure diversity, we kept the requirements minimal and encouraged many crowdworkers to participate in our experiments (we had 531 for the training set, 153 for the evaluation set).
>
> > _[concerns about lack of articulation of reasoning types]_
>
> We agree that it is not easy to formally define these different types of inference patterns or draw hard lines about how a system should behave. This is not our goal in this work; the analysis is merely meant to provide an illustration. Determining what kinds of systems will work well here (neurosymbolic approaches that can do explicit reasoning, purely end-to-end Transformer models, etc.) is an open question. Nevertheless, we think this dataset can be a useful benchmark for a wide variety of approaches that are exploring this question.
>
> > _we are never told that HP is good at riding on a broomstick_
>
> We agree; the example in the figure doesn’t necessarily outline all the reasoning needed for a system like a theorem prover to make the judgment. We will clarify this in the caption.
>
>
> > _it's not possible to use this dataset to examine how large-scale LMs will behave against real-world claims_
>
> We agree -- it would be great to have a dataset of more realistic claims. However, such claims are challenging to find at scale. Repositories of fact-checks on real entities involve significantly more background knowledge, including knowledge that may not be easily obtained. For the purposes of this work, which are primarily about understanding entity knowledge in NLP models, our assessment is that our human-authored claims are natural enough.

---

> > ### Comment · Reviewer_VoQi · 2021-09-29
> > **Thanks for the comprehensive response**
> >
> > I would like to thank the authors for their comprehensive response to my comments.
> >
> > Regarding my main concern, I see your point about the difficulties in controlling the reasoning categories. However, I maintain that for a dataset to be a useful benchmark a certain amount of control is needed - especially in a matter as subjective as commonsense reasoning. Of course, this is a matter of degree, and the even in its current form, the dataset might be useful to the community.

---

### Official Review · Reviewer_Stzx · 2021-09-20
**Interesting dataset bridging common sense and fact checking containing entities**

**Rating:** 7
**Confidence:** 3

**Strengths:**

Since entities usually posit a very specific and hard challenge for language models, this dataset can help to start bridging the gap between more artificial commonsense queries and naturalistic reasoning. The dataset itself would therefore be of value for the NLP community. The authors discuss potential ethical and social implications. The construction of the dataset itself is detailed and explained well. I especially appreciated the analysis on potential token-based artifacts models could pick up on (3.3 Dataset Artifacts).

**Weaknesses:**

I have two main concerns.
My first concern is with the authors' involvement in the analyses. It would be reassuring to see the performance and judgments of people that aren't directly involved in this work for assessing the human baseline since there is always a possibility that knowledge about the task provided to crowdworkers for instance can provide some advantage in evaluation performance. This holds similarly for the assessment of reasoning types in 3.3. Here, at least an estimate of interannotator agreement would already be informative.
Secondly, I'm wondering whether the gap between model performance and human performance is even theoretically possible to close. Specifically, models were trained at a specific point in time, which means they have no way of knowing certain celebrities that became important after they were trained, and presidents and other facts about the world change. Figure 4a and 4b already provide insights on model performance dependent on Wikipedia page views and entity types. It might be useful to include a measure for person and Wikipedia page creation date and relating that to the date of model training. There are probably other more sophisticated measures for dating claims and any would be useful to include.

**Additional Feedback:**

None

**Clarity:**

The paper is clearly written.

I only found that "Given how the contrast set was constructed, the fact that higher-capacity models work better suggests that having more entity knowledge is a key to doing better on this task." (line 293f) didn't follow from what I had read with respect to the creation of the contrast set. Was it specifically entities that were replaced to construct the contrast set?

typo:
- line 236: "models are ask"

**Correctness:**

The dataset creation and evaluation seem sound, keeping a potential implicit author bias in evaluation in mind (see Weaknesses).

**Documentation:**

The dataset is easily available and the code is well documented.

**Ethics:**

Since the reasoning samples were constructed by crowdworkers and, at least the training set, was not manually checked, they might contain instances that are problematic. The authors discuss this problem and clearly outline its limitations. The time-sensitive component, which is always an issue with knowledge bases and common sense datasets, should also be considered but is not a general worry.

**Relation To Prior Work:**

Prior work is cited and discussed throughout the paper.

**Summary And Contributions:**

The authors introduce CREAK, a dataset for investigating models' commonsense reasoning capabilities when involving entities. They assess the performance of two state-of-the-art pretrained language models against a strong SVM + TF-IDF and a human baseline. They find that current models fail to zero-shot generalize to CREAK. While fine-tuning on the dataset benefits performance, it still doesn't reach the human benchmark.

---

> ### Author Response · Authors · 2021-09-29
> **Author Response**
>
> We would like to thank the reviewer for the insightful comments!
>
> > _concern about the authors' involvement in the analyses_
>
> This is a good point. We conducted another round of human evaluation on 100 examples using a new group of annotators who were not involved in the project. On 100 development examples, they achieved 94% accuracy on consensus annotations while the authors achieved 99% (see Table 2). This performance estimate still leaves a large gap on the closed-book setting we focus on (a roughly 10 point gap from the best performing model).
>
> From inspection of the errors and discussion with the annotators, there were a few disagreements about where the line between true and false should be drawn, which could be addressed with additional annotator training and clearer instructions. For example, some annotators judged the claim “the dodo bird must have been a beautiful flightless bird that lived on Mauritius.” to be true, but some others marked as false since the word “beautiful” is subjective so it’s not *necessarily* true. Furthermore, there were two examples that were not subjective that annotators answered incorrectly, possibly due to performing the task quickly. We estimate that with training and a more careful job, these annotators would achieve closer to 97% accuracy.
>
>
> > _I’m wondering whether the gap between model performance and human performance is even theoretically possible to close [due to factors in RoBERTa training]_
>
> We agree that the temporal mismatch between LMs (training date) and datasets (creation date) can be a problem. RoBERTa was trained on the December 2018 EN Wikipedia dump, and we collected claims between June and August 2021. However, when we manually inspect 100 examples, none of the truth values would have changed in this time period and nearly all of the entities were already notable as of 2018. The only example that arguably relies on more recent knowledge is “Captain Marvel shows a woman using her swimming powers to save people”, since the popular Captain Marvel film was released in 2019. However, the character of Captain Marvel did exist before, so this information was within reach from the RoBERTa model. We will add discussions on temporal drift in any future version.

---

> > ### Comment · Reviewer_Stzx · 2021-09-29
> > **Revised rating**
> >
> > I would like to thank the authors for the additional analyses they performed. Since they addressed my main concerns, I revised my rating to reflect that. The dataset creation and quality control process seems sound which is why I recommend acceptance. However, my expertise in the field of common sense reasoning is not sufficient to assess how much the concerns of reviewer VoQi would affect the usability of the dataset.

---

### Official Review · Reviewer_aHYr · 2021-09-20
**A comprehensive dataset to study commonsense reasoning and entities**

**Rating:** 8
**Confidence:** 2
**Clarity:** The writing is clear and easy to follow.

**Strengths:**

1) The authors put serious consideration into the design and collection of this solid dataset,
including the diverse distribution of entities and their types, diverse set of reasoning types,
as well as the combination of commonsense reasoning and knowledge.
The procedure of data collection is very cautiously designed with rigorous quality control.
2) The paper presents comprehensive evaluation of the dataset with various different settings
and various systems for evaluation and comparison. Human performance is included for further
analysis.
3) The dataset is carefully constructed with room for future improvement, which is good
incentive for researchers to work on.

**Weaknesses:**

Overall, the dataset is good material for future research to be conducted on.
One minor issue might be that the study of bias in the dataset is still preliminary.

**Additional Feedback:**

N/A

**Correctness:**

The paper looks correct. The review is not aware of any major issues.


**Documentation:**

The authors provide detailed instructions to reproduce every experiment in the paper,


**Ethics:**

The authors include a paragraph on the intent that the paper shouldn't be used as general
fact-checking models, and the potential bias of fine-tuned models.
The review is not aware of any other ethical concerns.


**Relation To Prior Work:**

Literature review is comprehensive to the best knowledge of the reviewer.


**Summary And Contributions:**

This paper introduces a dataset that helps to evaluate two major topics in NLP models:
1) Entity understanding;
2) Commonsense inference.

These two topics are among the most important issues in NLP, and could potentially help
with many other tasks, including question answering, commonsense reasoning, knowledge probing, etc.

---

> ### Author Response · Authors · 2021-09-29
> **Author Response**
>
> We would like to thank the reviewer for the encouraging comments and feedback!
>
> > _the study of bias in the dataset is still preliminary_
>
> “Bias” here can mean two things. We investigate the word-level artifacts (i.e., spurious correlations between tokens and labels), and we show that CREAK contains relatively few artifacts, which are symptoms of “dataset bias.” There could also be social biases (gender, race etc.) in the dataset that are difficult to determine by quantitative tools and arise from biased judgements made by crowdworkers. To alleviate those biases, we manually inspected all evaluation examples and removed claims that can be seen as biased (e.g., subjective, offensive, harmful etc. -- see Appendix B, Table 5), which were not common in the dataset.

---

### Decision · Program_Chairs · 2021-10-09

**Decision:**

Accept

**Comment:**

The authors introduce CREAK, which is a benchmark for commonsense reasoning about entity knowledge, collecting true and false information about entities with crowdsourcing. The reviewers agree that this dataset would be of a valuable contribution for various NLP tasks. I strongly recommend acceptance of this paper.